# BCMA CAR-T: From Multiple Myeloma to Light-Chain Amyloidosis

**DOI:** 10.3390/curroncol32080418

**Published:** 2025-07-25

**Authors:** Ellen Lewis, Victor Hugo Jimenez-Zepeda

**Affiliations:** 1Arthur Child Comprehensive Cancer Centre, Calgary, AB T2N 4N2, Canada; ellen.lewis@ahs.ca; 2Division of Hematology, Department of Medicine, University of Calgary, Calgary, AB T2N 4N1, Canada; 3Arnie Charbonneau Cancer Research Institute, Calgary, AB T2N 4N2, Canada

**Keywords:** light-chain amyloidosis, BCMA, BCMA CAR-T, multiple myeloma

## Abstract

Light-chain (AL) amyloidosis is a rare and incurable clonal disease caused by misfolded toxic light-chain fibrils. As AL amyloidosis falls under the category of a plasma cell dyscrasia, treatment is often borrowed from that of similar diseases, such as multiple myeloma. To date, there are no FDA-approved therapies for patients with relapsed or refractory AL amyloidosis. A promising target for new treatments is BCMA, a protein involved in B-cell development. BCMA-directed therapies—such as CAR-T cells, bispecific antibodies, and antibody–drug conjugates—have shown success in treating relapsed multiple myeloma and are now being studied for AL amyloidosis. This review focuses on the current literature related to BCMA-targeted treatments, especially CAR-T therapy, in improving outcomes for AL amyloidosis patients.

## 1. Introduction

Light-chain amyloidosis (AL amyloidosis) is a rare clonal plasma cell disorder with an incidence of approximately 10 per million [1,2,3]. The disease arises when abnormal plasma cell clones produce amyloidogenic immunoglobulin light-chains that misfold into toxic fibrils [2]. These fibrils accumulate in extracellular spaces, causing structural damage and organ failure—typically systemic rather than localized [2]. Despite advances in early detection and treatment, AL amyloidosis remains incurable [2,4].

Due to its rarity and patients’ frequent exclusion from clinical trials—often because of significant organ dysfunction—AL amyloidosis treatments are commonly adapted from strategies used in other plasma cell disorders, such as multiple myeloma (MM) [5]. Although disease-specific trials have increased over the past decade and regional clinical practice guidelines exist [6], treatment decisions in AL amyloidosis—especially in the relapsed/refractory (RR) setting—remain uncertain [5,7]. While data on B-cell maturation antigen (BCMA)-targeted chimeric antigen receptor T-cell (CAR-T) therapy in RR AL amyloidosis is minimal, understanding this emerging treatment avenue is increasingly important. This paper provides a historical overview of AL amyloidosis treatment, discusses BCMA as an immunotherapy target, reviews current data on CAR-T use in AL amyloidosis, and compares its efficacy and tolerability to that in RRMM.

## 2. Historical Treatment of AL Amyloidosis

Historically, initial treatment for AL amyloidosis involved the use of melphalan in combination with a steroid such as dexamethasone or prednisone [8,9]. This approach, adapted from MM treatment protocols, resulted in a median overall survival (OS) of 12–18 months for AL patients [10]. In the 1990s, autologous stem-cell transplantation (ASCT) was introduced [11]. Although ASCT led to improved organ responses in some individuals [12], a systematic review published in 2009 concluded that ASCT provided no clear survival benefit over conventional chemotherapy alone [13]. It is important to note that this 2009 review was conducted before the widespread implementation of cardiac biomarkers, which have since enhanced risk stratification and reinforced the necessity of careful patient selection for ASCT [14]. While high-dose chemotherapy followed by ASCT remains a therapeutic option, only a limited subset of patients typically meets the criteria for this approach [5,14,15].

In the early 2010s, treatment shifted to bortezomib-based therapies, with cyclophosphamide, bortezomib, and dexamethasone (CyBorD) becoming the most widely used regimen. Studies demonstrated that CyBorD led to improved hematologic and organ responses, as well as better OS compared to melphalan-based regimens [16,17,18,19]. Despite its continued use as standard-of-care in some regions, common cytogenetic abnormalities—especially t(11;14)—are associated with poor or absent responses to bortezomib in some patients [20].

The phase III ANDROMEDA trial, which added daratumumab to CyBorD (Dara-CyBorD), marked a key advancement in frontline AL amyloidosis therapy. At a median follow-up of 11.4 months, Dara-CyBorD showed higher rates of hematologic complete response (CR) (53% vs. 18%), cardiac response (42% vs. 22%), and renal response (53% vs. 24%) compared to CyBorD alone [21,22]. Dara-CyBorD remains the only FDA-approved first-line treatment for AL amyloidosis [23], but optimal approaches for subsequent therapy lines remain unclear. Options for daratumumab-refractory patients include carfilzomib, ixazomib, or immune modulator-based therapies, although these agents may yield additional toxicities and less durable responses [24,25,26,27]. Additionally, venetoclax, under investigation for patients with t(11;14), has shown promising responses and tolerability in small retrospective studies [28,29].

## 3. BCMA Targeted Therapy

BCMA has emerged as a promising target in the management of AL amyloidosis, particularly in patients with concurrent plasma cell disorders such as RRMM [30,31]. BCMA, a member of the tumor necrosis factor (TNF) receptor superfamily, is primarily expressed on terminally differentiated B cells, especially plasma cells [30,31,32]. It binds the ligands APRIL and BAFF, both of which support the survival and proliferation of plasma cells [30]. BCMA targeted therapies include antibody–drug conjugates, bispecific antibodies, trispecific antibodies, and NK cell engagers. Although CAR-T cell therapy is also included under the umbrella of BCMA therapies, it will be discussed separately.

### 3.1. Antibody Drug Conjugates

Belantamab is an IgG1 antibody–drug conjugate [33] which received approval for RRMM based on results from the DREAMM-2 clinical trial [34,35]. The results from the DREAMM-2 trial showed an overall response rate (ORR) of 32% with a median response duration of 12.5 months in a group of heavily pre-treated RRMM patients [34,35]. Belantamab monotherapy was subsequently evaluated in RR AL amyloidosis patients, with a phase-2 clinical trial by the European Myeloma Network, which reported a hematologic ORR of approximately 50% [36,37]. Similarly, Khwaja et al. reported a hematologic ORR nearing 70% in RR AL amyloidosis patients using belantamab monotherapy [38]. Despite these promising response rates, belantamab is associated with significant ocular toxicity, raising concerns regarding its overall tolerability [34,35,36,37,38].

### 3.2. Bispecific and Trispecific Antibodies

Teclistamab is a bispecific IgG4 antibody that engages T-cells to target BCMA. Its approval for RRMM was based on data from the MajesTEC trial [39,40]. Patients in the MajesTEC trial were heavily pre-treated with a median of five lines of prior therapy; promisingly, an ORR of 63% was noted in the teclistamab cohort [39]. Teclistamab was subsequently studied in patients with coexisting RRMM and AL amyloidosis. In a small case series, all eight patients achieved at least a very good partial response (VGPR), with responses sustained at a median follow-up of 13 months in 5/8 patients [41]. Additionally, Forgeard et al. reported a retrospective case series involving 17 AL amyloidosis patients treated with teclistamab, 10 of whom had concurrent MM [42]. Among these, 15 patients achieved a hematologic VGPR or better, and five attained an organ response [42].

Elranatamab, another approved bispecific T-cell engager developed for RRMM, received approval based on findings from the MagnetisMM-3 study [43]. The MagnetisMM-3 trial noted an ORR of 61% in a cohort of heavily pre-treated RRMM patients with a median of five prior lines of therapy [43]. In 2024, Vianna et al. published a retrospective analysis involving three patients with daratumumab-refractory AL amyloidosis; all achieved a complete hematologic response (CR) following elranatamab therapy, with responses sustained at a median follow-up of 141 days [44].

Two additional bispecific antibodies, ABBV-383 (NCT06158854) [45] and Linvoseltamab (NCT06292780) [46] are currently recruiting in the RR AL amyloidosis setting. The trispecific antibody JNJ-79635322 (NCT05652335) [47] is also currently recruiting for both RRMM and RR AL amyloidosis.

### 3.3. NK Cell Engagers

The NK-cell engager SER 445514 (NCT05839626) [48], which is a first-in-human phase 1 study in RRMM and RR AL amyloidosis setting, is active and has completed recruiting. This study was initiated in May of 2023 with anticipated results in 2028.

## 4. BCMA CAR-T Therapy

CAR-T is a treatment that consists of leukopharesing a patient’s own T-cells and modifying them to contain a chimeric antigen receptor that targets antigens expressed on tumor cells resulting in T-cell mediated cell death [49]. In vitro analysis has shown that elevated BCMA on malignant plasma cells aids in cancer survival pathways [50]; however, BCMA expression varies between different plasma cell dyscrasias. When comparing BCMA expression on bone marrow samples for patients diagnosed with MM and AL amyloidosis, BCMA expression in AL amyloidosis was lower compared to MM [51]. This was replicated in a pre-clinical study of RR AL amyloidosis patients, whose percentage of BCMA-positive cells was similar to that of MM, but the intensity of expression was weaker [52]. When developing CAR-T HB10101, Krif-Erendfeld and colleagues (2022) noted that the mean fluorescence intensity of BMCA on AL amyloid cells averaged 1.9 (n = 18) when compared to 3.8 in MM cells (n = 39); however, this observation did not appear to impact efficacy [52]. When cocultured in vitro with BCMA CAR-T, pathologic AL plasma cells were markedly cleared [52].

## 5. BMCA CAR-T Efficacy

To evaluate the efficacy of BCMA CAR-T therapy in AL amyloidosis, it is essential to contextualize the patient populations reported in the current literature (Table 1). A total of 34 patients with AL amyloidosis having received a BCMA CAR-T product and have been reported to date [53,54,55,56,57]. The data on these patients can be obtained from three total case(s), one trial, and data from an in-term clinical trial (Table 1). Thirteen of these patients had concurrent RRMM, 21 patients studied in the trials by Lebel et al. and Landau et al. had isolated AL amyloidosis [55,57]. Regardless of MM status, most patients were heavily pre-treated, with a median of four prior lines of therapy [55,57]. AL amyloidosis disease staging was predominantly early, with 76.6% (n = 23) of the 30 evaluable patients classified as Mayo stage I–II, 20% (n = 6) as stage III, and a single patient as stage IV [53,54,55,56,57]. Organ involvement was diverse: cardiac infiltration was noted in 21 patients, followed by soft tissue (n = 11), gastrointestinal (GI; n = 7), liver (n = 6), peripheral nervous system (PNS; n = 6), renal (n = 6), pulmonary (n = 1), and bladder (n = 1) involvement [53,54,55,56,57]. Among the 34 AL amyloidosis patients treated with BCMA-targeted CAR-T therapies, 23 received NXC-201, 7 received ide-cel, 3 were treated with cilta-cel, and 1 received ARI0002h [53,54,55,56,57].

### 5.1. Idecabtagene Vicleucel (Ide-Cel)

Idecabtagene vicleucel (ide-cel) is a second-generation CAR-T product designed to target BCMA through a single infusion of autologous T cells [49]. These cells are genetically modified and re-infused following lymphodepletion with cyclophosphamide and fludarabine [49]. Originally approved for the treatment of RRMM in patients who have received at least three prior lines of therapy, ide-cel has shown consistent efficacy in this setting [58,59]. The phase II KarMMA-2 study demonstrated a response rate of 73.4% in 128 heavily pre-treated patients, with 31.3% achieving a complete response (CR) (*p* < 0.0001) [58]. This was followed by the phase III KarMMA-3 randomized controlled trial, where 386 patients showed a median progression-free survival (PFS) of 13.3 months in the ide-cel group versus 4.4 months in those on standard therapy (*p* < 0.001) [59].

Existing data on Ide-cel usage for AL amyloidosis patients is limited to small case series but overall, shows promise. In 2023 Das et al. reported a case of one patient with concurrent AL amyloidosis and RRMM who received ide-cel [56]. The patient was a 62-year-old with R-ISS stage II MM with concurrent diagnosis of Kappa AL amyloidosis discovered on gastric biopsy [56]. The patient was daratumumab-refractory and had received five prior lines of therapy including an ASCT [56]. They went on to develop renal disease and cardiac involvement, conferring a Mayo stage II amyloidosis [56]. Ide-cel administration was complicated by cytopenias, the patient received CNS prophylaxis and did not develop ICANS or CRS [56]. At 30-day re-evaluation the patient attained a VGPR, was MRD negative, with a >30% decrease in NT-ProBNP. VGPR and stable organ function persisted at 9 months [56].

In 2024 Goel et al. reported retrospective cases of eight patients with RRMM and concurrent AL amyloidosis who received ide-cel (n = 6) or cilta-cel (n = 2) [54]. Two of the eight had cardiac involvement, one with renal, one with GI, and four with soft tissue. Patients had a median of eight prior lines of therapy (range 6–11), all eight were daratumumab-refractory, and six had received prior ASCT [54]. Treatments were complicated by CRS in six patients, cytopenias in seven patients, one developed ICANS, and three patients developed viral infections [54]. At the median follow-up of 11 months (range 5.6–26.4), three patients attained a hematologic CR, two achieved a hematologic VGPR, and the final three patients were deemed non-evaluable as they did not have measurable serum free light chains at the time of the CAR-T infusion [54]. Time to best response was 43 days (range 20–46) [54].

### 5.2. Ciltacabtagene Autoleucel (Cilta-Cel)

Ciltacabtagene autoleucel (cilta-cel) is another second-generation CAR-T therapy with a dual-antibody construct targeting BCMA [60]. Like ide-cel, it involves a single infusion of autologous T cells after lymphodepletion [60]. Data leading to its approval included the CARTITUDE-1 trial (MMY2001) trial of 97 heavily pre-treated RRMM patients (median six prior lines of therapy) [61,62]. The overall response rate (ORR) was 97% (n = 94), with 67% (n = 65) of the studied patients achieving a stringent CR (sCR). Twelve-month progression-free survival (PFS) was 77% and the OS was 89% (Berdeja, 2021) [62].

Similarly to Ide-cel, data on cilta-cel use in AL amyloidosis patients is limited. In addition to the 2 AL amyloidosis patient cases described by Goel et al. (2024) [54] case series on second-generation BMCA CAR-T therapy in amyloidosis (summarized above), Das et al. provides one additional patient case for review [56]. In 2023, Das reported a case of a 33-year-old with concurrent RRMM and Mayo 2012 stage IV AL amyloidosis with cardiac involvement [56]. They were daratumumab-refractory and had 4-prior lines of therapy [56]. The patient received cilta-cel with dexamethasone for CRS prophylaxis [56]. Treatment was complicated by grade III CRS requiring ICU support, and cytopenias [56]. At the 30-day re-evaluation the patient had new lytic lesions on PET-CT but was minimal residual disease (MRD)-negative [56]. At the 9-month follow-up patient remained MRD negative, in an sCR, with a 30% reduction in NT-Pro-BNP, and complete resolution of previous FDG-avid lytic lesions [56].

### 5.3. Cesnicabtagene Autoleucel (ARI0002h)

ARI0002h is a second-generation humanized BCMA CAR-T and is currently being studied in the RRMM populations in the CARTBCMA-HCP-01 trial [63]. One 60-year-old patient in the trial had R-ISS Stage II MM treated previously with ASCT, carfilzomib, and daratumumab, and subsequently developed lambda AL amyloidosis with renal involvement while on daratumumab monotherapy [53]. The patient was treated with AR10002h on a compassionate basis as AL amyloidosis patients were excluded from the CARTBCMA-HCP-01 trial [53]. The AR10002h treatment was complicated by cytopenias, mild CRS, and infections [53]. At one-month follow-up, the patient attained a hematologic PR and a 33% decrease in proteinuria [53]. At the 6-month follow-up they attained an sCR, were MRD negative and had a 55% decrease in proteinuria which persisted at the 12-month follow-up [53].

### 5.4. NXC-201 (Formerly HBI0101)

NXC-201, is a newer BCMA CAR-T therapy, again initially designed for use in the RRMM setting [55]. NXC-201 is a heavy-chain connected to a light-chain of which both are derived from the C11D5.3 antibody [52]. When developing NXC-201, pre-clinical analysis of 24 patients showed that the percentage of BCMA-positive cells was similar between MM and AL amyloidosis patients; however, the intensity of expression was diminished for AL amyloid plasma cells [52]. Despite the diminished BCMA intensity of AL amyloidosis cells, NXC-201 demonstrated robust clearing of pathologic amyloid cells in vitro [52].

Subsequently, NXC-201 was studied via the NEXICART phase I trial, which consisted of predominantly RRMM patients in addition to 16 AL amyloidosis patients (2 with concurrent MM) which were analyzed in a separate subgroup [55]. The AL amyloidosis sample consisted of 16 heavily pre-treated patients with a median four prior lines of therapy (range 3–10) [55]. The majority had cardiac (13/16) or renal (11/16) involvement followed by soft tissue (6/16), peripheral nervous system (6/16), liver (6/16), GI (5/16), and lung (1/16) involvement [55]. A total of five patients had Mayo stage III–IV disease [55]. Treatment with NXC-201 was complicated CRS in much of the cohort and all patients developed short-lived cytopenias [55]. Infections were also common, and one patient died in a CR from a COVID-19 infection [55]. Overall, 12/16 patients attained a CR, 2/16 VGPR, 1/16 PR, and 1/16 had no response [55]. Time to best response was 17 days and 9/14 evaluable patients achieved MRD negativity [55]. Organ responses were seen in 8/13 evaluable patients, mostly cardiac (7/9 patients) and renal (2/6) [55]. At a median follow-up of 8.4 months, seven patients had died, three while in CR/VGPR, and four died during relapse [55].

NEXICART-2 Phase 1b trial (NCT06097832), which specifically is evaluating CAR-T therapy in RR AL amyloidosis patients is currently enrolling [64]. Data from the NEXICART-2 trial was presented at the American Society of Clinical Oncology conference in May of 2025 and an abstract from this conference shared interim data on the 7 RR AL amyloid patients who have received NXC-201 CAR-T to date [57]. The initial cohort of AL amyloidosis patients had a median of four (range 2–9) prior lines of therapy, all were RR to bortezomib and anti-CD-38 therapy, and four patients had a prior ASCT [57]. Patients had mainly cardiac involvement (4/7; 53%) with two patients having Mayo stage I disease, four having stage II disease, and one patient having stage IIIa disease [57]. CRS was a main complication affecting 5/7 patents, all were low grade (I–II) [57]. No neurotoxicity was observed [57]. Grade III–IV neutropenia was seen in five patients, with no episodes of febrile neutropenia or treatment-related infections at a median follow-up of 97 days (range 7–209) [57]. ORR was 100% and all seven patients attained a VGPR or better, while five out of the seven evaluable patients were MRD negative [57]. One patient with renal disease was noted to have an organ response [57]. The NEXICART-2 trial remains ongoing with the aim to enroll 40 patients [64].

## 6. Tolerability of BCMA CAR-T in AL Amyloidosis

The safety and tolerability of the various CAR-T products were similar across the 34 patients summarized in the available data [53,54,55,56,57]. See Figure 1. When compared to adverse events and tolerability of BCMA CAR-T in the RRMM setting, outcomes for AL amyloidosis patients were similar. The most common and frequently reported adverse reactions and side effects in the literature were CRS, ICANS, cytopenias, and infections [53,54,55,56,57].

### 6.1. Cytokine Release Syndrome (CRS)

Cytokine release syndrome (CRS) is a well-documented and frequently observed complication following BCMA-targeted CAR-T cell therapy [53,54,55,56,57,58,62,64,65]. It represents a systemic inflammatory response driven by massive cytokine secretion, secondary to T-cell activation [64,65]. CRS can be graded; the most common grading system is by Lee et al., 2019 [66]. Clinical presentations may vary from mild symptoms such as headache, fever, and malaise to more severe manifestations like multi-organ failure, often requiring vasopressors and supplemental oxygen [64,65,66].

When looking at data from larger trials on CAR-T use in RRMM patients, CRS occurred in 84% (n = 107) of ide-cel patients in the KarMMA-2 trial, with only 5% (n = 7) patients developing grade II–IV CRS [58]. Additionally, CRS developed in 95% (n = 92) of the cilta-cel patients treated in the CARTITUDE-1 trial, with 4% (n = 4) being grade III–V [62].

In patients with AL amyloidosis, similar CRS trends are observed. Among the 34 patients reported in the literature treated with BCMA CAR-T therapy, 28 experienced CRS [53,54,55,56,57]. Das et al. (2023) described a case involving a grade III CRS event on day six following cilta-cel infusion, despite prophylactic dexamethasone [56]. This patient required intensive care, vasopressors, and tocilizumab but recovered within 24 h [56]. Goel et al. (2024) reported CRS in 6/8 patients treated with either ide-cel or cilta-cel, all of which were grade II or less [54]. Notably, none of these patients received CRS prophylaxis, although four went on to require tocilizumab [54]. Oliver-Caldes et al. (2021) reported the first known AL amyloidosis case treated with ARI0002h, in which the patient developed grade I CRS on day eight [53]. The episode resolved spontaneously within 48 h without intervention [53].

In contrast, CRS following NXC-201 therapy appeared more rapid onset, with 19/23 patients experienced CRS within 1 to 3 days post-infusion [55,57]. These cases included three grade III events, nine grade II, and seven grade I events [55,57]. Seventeen patients required tocilizumab, and three patients in the NETICART-1 trial were treated with corticosteroids [55,57]. Symptom duration resolved with hours-short days [55,57].

### 6.2. Immune Effector Cell-Associated Neurotoxicity Syndrome (ICANS)

ICANS, commonly known as neurotoxicity, is another potential complication of CAR-T therapy. As previously hypothesized, its pathophysiology is believed to involve cytokine-mediated endothelial activation and disruption of the blood–brain barrier [67,68]. Presentations can range from mild cognitive changes and headaches to severe cerebral edema and coma [67,68].

In large RRMM studies, ICANS was observed in 21% of cilta-cel recipients in the CARTITUDE-1 trial, with 9% experiencing grade III–V events [62]. Similarly, the KarMMA-2 trial reported neurotoxicity in 18% of ide-cel patients, with 3% of those classified as grade III or higher [58]. However, in the AL amyloidosis patients summarized in the current literature, ICANS has been infrequently reported. Among the 34 documented patients, only one case of grade III ICANS was noted, following ide-cel infusion [54]. While vigilance is warranted, current data suggest ICANS may occur less frequently in the AL amyloidosis population.

### 6.3. Cytopenias

Cytopenias, whether transient or prolonged, are among the most common hematologic adverse events following CAR-T therapy. The etiology is likely multifactorial, involving lymphodepleting chemotherapy, extensive prior treatments, baseline cytopenias, and prior ASCT [69,70]. In CARTITUDE-1 trial, neutropenia, anemia, and thrombocytopenia were reported in 96% (n = 93), 81% (n = 79), and 79% (n = 77) of cilta-cel patients, respectively, with most counts normalizing by day 30 [62]. The KarMMA-2 trial noted neutropenia in 91% (n = 117) of ide-cel patients, anemia in 70% (n = 89), and thrombocytopenia in 63% (n = 81) [58]. Most cytopenias resolved within one month; however, among those with prolonged cytopenias, the median recovery time to mild-grade neutropenia and thrombocytopenia was 1.9 and 2.1 months, respectively [58].

Similarly to RRMM patients, Cytopenias were the most common development amongst the 34 AL amyloidosis patients studied. The most notable being neutropenia, which was seen in 17/23 of the NXC-201 patients in the NexiCART-1 and 2 trials, 5/8 patients summarized by Goel, 2/2 patients summarized by Das, and the single patient who received AR10002h [53,54,55,56,57]. Most of the neutropenia episodes developed were grade IV and most recovered within 1 month of BCMA CAR-T administration [53,54,55,56,57]. A few patients reported went on to require additional support, including the 5/8 patents summarized by Goel at al. who required G-CSF, and 1/8 required a stem cell boost for cytopenia management [54].

Thrombocytopenia was also frequently reported, with 9/16 NXC-201 patients in the NexiCART-1 trial developing short-lived grade I–II thrombocytopenia, one cilta-cel patient with grade III thrombocytopenia, and the patient who received AR10002h developing grade III thrombocytopenia which improved to Grade I and persisted past 12 months [53,54,55,56]. Two/eight patients described by Goel et al. (2023) developed grade I–II thrombocytopenia within the first 30 days, one which required a thrombopoietin receptor agonist [54]. Thrombocytopenia was not mentioned in the abstract of the NEXTCART-2 trial [57]. In the reports of patients who received ide-cel and citla-cel, grade I-II anemia was reported in 3/8 patients described by Goel et al. and both patents described by Das et al. [54,56] Das et al. reported that both patents (1 cilta-cel and 1 ide-cel) continued with grade I anemia when last evaluated [56]. Twelve of the sixteen patients who received NXC-201in the NEXTCART-1 trial developed anemia, 5/16 being grade III–IV, with all but two resolving within the first month [55].

### 6.4. Infections

The development of early and late infections is a common concern in patients treated with CAR-T therapy. Patients are both at risk due to the inherent neutropenia that develops post-leukoreductive chemotherapy prior to CAR-T and remain at risk because of associated hypogammaglobulinemia secondary to the CAR-T therapy itself [64]. Infections amongst RRMM patients treated with cilta-cel were reported in 58% (n = 56) patients and 20% of those were grade III–IV [62]. In the KarMM-2 study, infection occurred in 68% of patients (n = 88), and 22% were noted to be grade III–IV [58].

When looking at the BCMA CAR-T AL amyloidosis patients, the AR10002h patient went on to develop two separate infections within the first few months [53]. The first was COVID-19 infection which resolved without issue and the second being a BK virus-associated hemorrhagic cystitis requiring frequent intervention and IV immunoglobulins [53]. In terms of early infections, 9/16 patients who received NXC-201 (via NexiCART-1 trial) developed an infection within the first 28 days post CAR-T [55]. Six total patients had grade III infections (five febrile neutropenia, one pneumonia); 2/9 had grade I–II respiratory infections, and 1/9 had early CMV [55]. One year post NexiCART-1 NXC-201 CAR-T therapy, 3/16 patients required IVIG treatment and 7/16 developed late-term infections, one being Grade V COVID-19 infection [55]. NexiCART-2 trial in-term data analysis commented that there were no treatment-related infections in the seven patients summarized [57]. Goel et al. described respiratory infections developing in 3/8 patients, only one being above a grade III [54].

## 7. Discussion

AL amyloidosis remains a complex and challenging disease to treat. Clinicians often adapt therapeutic strategies and pharmacological advancements from the MM setting, owing to the shared pathophysiological basis of these clonal plasma cell disorders [5,7]. The success of BCMA-directed CAR-T therapy in RRMM has influenced the exploration of this treatment modality in AL amyloidosis. Encouragingly, the limited clinical experience to date—including several case reports and small cohort studies—suggests both promising efficacy and a manageable safety profile for BCMA CAR-T in this population [53,54,55,56,57].

Despite evidence suggesting that BCMA expression is lower in plasma cells derived from AL amyloidosis compared to MM [50,52], this did not appear to negatively impact the therapeutic efficacy of BCMA-targeted CAR-T therapy. Of the 31 evaluable AL amyloidosis patients summarized in the 5 case (s)/trials, a remarkable 93.5% (n = 29) achieved a hematologic response of VGPR or better: 22.5% (n = 7) reached VGPR, 64.5% (n = 20) achieved a CR, and 8.3% (n = 2) attained an sCR [53,54,55,56,57]. MRD negativity was documented in 68% (n = 17) of the 25 evaluable patients [53,54,55,56,57]. Only one patient receiving NXC-201 in the NEXICART-1 trial failed to respond, while another achieved only a PR [55].

Due to frequent systemic organ involvement and functional impairment in AL amyloidosis, treatment decisions are typically made with heightened caution [5,7]. While most patients in the available literature had early-stage disease, the incidence and severity of adverse events were largely comparable to those observed in RRMM trials [53,54,55,56,57,58,62]. The rates and grading of CRS were similar, with only one patient—diagnosed with stage IV AL amyloidosis—requiring intensive care for CRS management [56]. Importantly, ICANS was observed in only a single patient, representing a lower frequency than typically reported in MM populations [54,58,62]. Infection rates in the evaluable patients were also comparable to those reported in RRMM cohorts, though one patient, who was in CR, succumbed to COVID-19 [55]. Cytopenias were common but generally short-lived [53,54,55,56,57]. The lower incidence of prolonged cytopenias in patients with isolated AL amyloidosis may reflect better baseline marrow reserves and lower clonal plasma cell burden compared to those with concurrent MM [55].

Given that BCMA CAR-T is a newer treatment consideration for AL amyloidosis patients, late-onset events and survivorship care have not been fully uncovered in the current literature. Prolonged cytopenias and late-onset infections have been described in a few AL amyloidosis patients discussed, but more data is needed to understand the implications. Reviews on CAR-T therapy use have described late effects including secondary malignancies, psychosocial impacts, fertility concerns, organ dysfunction, and autoimmunity [71]. It is too early to fully understand the longer-term impacts of CAR-T therapy for the very nuanced population of AL amyloidosis patients and as such future research is needed in this area.

A notable limitation across the available studies is the relatively short duration of follow-up, with most reports limited to approximately one year [53,54,55,56,57]. Nonetheless, these early findings offer significant hope for a patient population historically marked by limited treatment options and poor outcomes. These data support the continued practice of adapting successful MM-based therapies for AL amyloidosis, including the use of BCMA-directed CAR-T therapy.

## 8. Conclusions and Future Directions

Over recent decades, significant progress has been made in the diagnosis and treatment of AL amyloidosis, with improved survival outcomes driven by therapeutic advancements. Given the rarity of the disease and the frequent presence of organ dysfunction, treatment regimens are often adapted from the multiple myeloma setting. The development and approval of BCMA-directed CAR-T therapies represent a major milestone in RRMM management, and their application to AL amyloidosis is a logical and promising next step.

This paper has summarized the currently available literature on the use of BCMA-targeted CAR-T therapy in AL amyloidosis, comparing outcomes to the more robust data available in RRMM. Early findings suggest that the efficacy and safety outcomes in AL amyloidosis are comparable to those in MM, with high ORR and acceptable toxicity profiles.

Future research should aim to include larger cohorts, particularly focusing on patients with AL amyloidosis without concurrent MM. In addition, extended follow-up is essential to better assess the long-term durability of response and late-onset toxicities. With continued investigation, BCMA-directed CAR-T therapy may become a cornerstone of treatment for relapsed and refractory AL amyloidosis.

## Figures and Tables

**Figure 1 curroncol-32-00418-f001:**
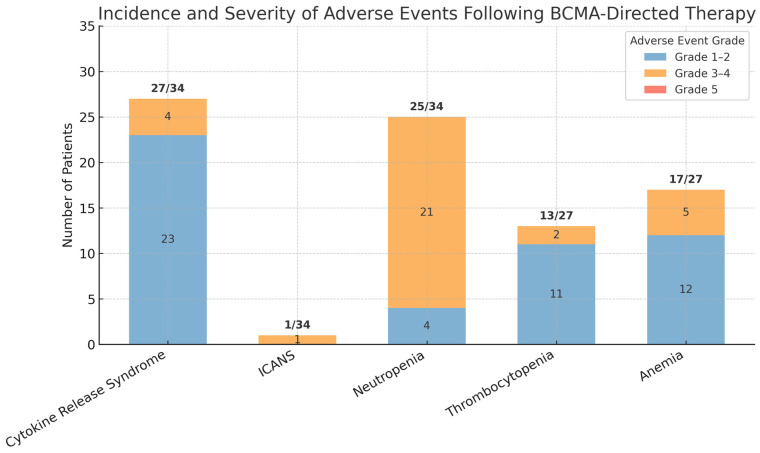
Incidence of treatment-related adverse events (by grade) in patients receiving BCMA-targeted therapy amongst summarized studies. Grades are categorized according to CTCAE v5.0: Grade 1–2 (mild/moderate), Grade 3–4 (severe/life-threatening), and Grade 5 (death). A total of 34 patients were evaluable for CRS, ICANS, and neutropenia. Total of 27 patients evaluable for thrombocytopenia and anemia.

**Table 1 curroncol-32-00418-t001:** Summary of BCMA CAR-T studies in AL Amyloidosis.

Data Source	Oliver-Caldes et al., 2021 [53]	Das et al., 2023 [56]	Goel et al., 2024 [54]	Lebel et al., 2024 [55] NexiCART-1	Landau et al., 2025 [57] ** NexiCART-2 Trial Ongoing **
BCMA CAR-T Product	** *ARI0002h* **	Ide-cel (1) and Cilta-cel (1)	Ide-cel (6) and Cilta-cel (2)	NCX-201 (Formerly ***HBI010****)*	NCX-201
Number of Patient(s)	n = 1	n = 2	n = 8	n = 16	n = 7
Patient(s) Age Mean (range)	Early 60s	Patient 1 (Ide-cel)—62 Patient 2 (Cilta-cel)—33	70.5 (range 56–75)	64 (range 55–82)	66 (range 56–82)
Diagnosis and Stage	Mayo stage II AL amyloidosis with concurrent R-ISS stage II IgA-lambda MM	Patient 1—Mayo stage II Kappa AL amyloid with concurrent R-ISS stage II MM Patient 2—Mayo stage IV Lambda AL amyloidosis with concurrent MM	AL amyloidosis with concurrent MM Mayo AL amyloidosis stage I (n = 1) II (n = 3) n/a (n = 4) R-ISS stage I (n = 2) II (n = 5) III (n = 0) n/a (n = 1)	AL amyloidosis (n = 14). AL amyloidosis with concurrent MM (n = 2) Mayo staging I-II (n = 11) IIIa (n = 4) IIIb (n = 1)	AL amyloidosis Mayo staging I (n = 2) II (n = 4) IIIa (n = 1)
Organ Involvement	Renal and Bladder	Patient 1—Gastric and Cardiac Patient 2—Cardiac, soft tissue,	Cardiac (n = 2) Renal (n = 1) GI (n = 1) Soft tissue (n = 4)	Heart (n = 13) Renal (n = 11) Soft tissue (n = 6) PNS (n = 6) Liver (n = 6) GI (n = 5) Lung (n = 1)	Heart (n = 4) Renal (n = 2)
Prior Therapy	3	4	8 (range 6–11)	4 (range 3–10)	4 (range 2–9)
Best Hematologic Response	ORR = 100% sCR. MRD negative	ORR= 100% Patient 1—VGPR. MRD negative Patient 2—sCR. MRD negative	ORR= 100% in 5 evaluable CR (n = 3) VGPR (n = 2) Not-evaluable (n = 3)	ORR = 94% CR (n = 12) VGPR (n = 2) PR (n = 1) No response (n = 1)	ORR = 100% VGPR/CR (n = 7). MRD negative (n = 5)
Organ Response	55% decrease in proteinuria	Patient 1—>30% reduction NT-ProBNP Patient 2—>30% reduction in NT-ProBNP	n/a	Any organ response 62% (8/13) Cardiac 78% (7/9) Renal 33% (2/6) Hepatic 20% (1/5)	Renal response obtained (n = 1)

n/a: non available; **: study ongoing and enrolling.

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
