# Peer review of "BCMA CAR-T: From Multiple Myeloma to Light-Chain Amyloidosis"

_curroncol, 2025, doi:10.3390/curroncol32080418_

Round 1

Reviewer 1 Report

Comments and Suggestions for Authors

This is a very good paper about an hot-topic as the treatment of RR AL Amyloodosis and the role of the new TCR in this field.

Besides some minor error in the transcription some suggestionS.

  1. Can you explain briefly the current options for the RR AL Amyloidosis treatment?
  2. May you correct Majestec with Majestec-1? May you cite DREAMM -7 and 8?
  3. Can you briefly explain the mechanism of action of the ADCs and Bispecific/Trispecific Antibodies?
  4. About the discussion can you provide your suggestions/comments about pros and con of TCR in AL Amyloidosis particularly about the ideal patients and the cardiac staging?  What is the major obstacle to the access to CAR-T or TCR for this population?

Author Response

  1. Can you explain briefly the current options for the RR AL Amyloidosis treatment? Thanks for the comment, in section 2, we have revised the relapsed refractory treatment options and the considerations to take while deciding which drug to use.
  2. May you correct Majestec with Majestec-1? This has been corrected in the revised version.
  3.  May you cite DREAMM -7 and 8? Thanks for the suggestion. We have added these studies to our revised version.
  4. Can you briefly explain the mechanism of action of the ADCs and Bispecific/Trispecific Antibodies? Thanks for the note, This has been added to the revised version of the manuscript.
  5. About the discussion can you provide your suggestions/comments about pros and con of TCR in AL Amyloidosis particularly about the ideal patients and the cardiac staging?  What is the major obstacle to the access to CAR-T or TCR for this population? Thanks for the suggestion. We have added to the final paragraph of the discussion section and made a separate future directions section addressing CAR-T access.

Title

I suggest that you replace “BCMA CAR-T” with “BCMA-targeted, or BCMA-directed therapy” since the review does not just describe CAR-T cell therapy. Thanks for the comment.  We have changed the title has been made as suggested

Paragraph 2

Line 69: are associated “with inferior response rates and survival outcomes” The statement here has been attenuated as suggested.

Line 71: “evaluated the addition of daratumumab to the standard of care…” Thanks for the note. We have changed this comment as suggested.

Paragraph 3

Line 94: Add the median prior lines of therapy. Thanks for the note. This has been added in the revised version of the manuscript.

Line 100: I do not fully agree with this statement, as ocular toxicity has been demonstrated to be manageable with schedule modifications with no significant impact on outcomes. Thanks for the comment, we have  changed/added a comment about this as suggested.

Line 113: “developed approved in RRMM” The explanation of RRMM has been provided in the revised version

Line 125: Explicate that the therapeutic target is BCMA. Thanks for the note. This has been corrected

Paragraph 5

Line 144: BMCA BCMA. Thanks for highlighting this typo, this has been corrected in the revised version of the manuscript

Line 146: and

Line 141: “from 3 total case(s)?: do you mean “from 3 case reports”? Thanks for the note. This has been clarified in the revised version.

Line 148: thirteen 13 (use the number, in line with the other data reported). Thanks for the comment, we have modified this to include Thirteen instead.

Line 154: cardiac infiltration involvement. Can you specify how many patients had advanced cardiac involvement? Thanks for the note. Unfortunately, we can’t really do this as some of the reports didn’t explicitly note degree of cardiac involvement.

Line 149: NCX. This has been fixed in the revised version

Line 170: discovered diagnosed on gastric biopsy. Thanks for the note. We have edited this line as suggested.

Line 170: They? Thanks for the note. We have changed to read “the patient”

Line 180: Can you grade CRS, ICANS and infections? Thanks for the note, This has been added in the revised version of the maniuscript

Lines 200-202: Unclear, please rephrase. Thanks for the note, this has been modified in the revised version of the manuscript

Line 201: MRD-negative in the bone marrow 2 . This has been corrected

Line 242: Although response rates are quite impressive, the mortality rate is high as well. Is the cause of death reported in the study (infections? organ failure).Thanks for the comment. The study described here showed indeed a higher mortality likely combination of infections and organ involvement. Unfortunately, not much more detail into the deaths is described in the paper.

Paragraph 6

Line 265: grade II-IV grade III-IV CRS? This has been modified in the revised version

Line 296: Immune-mediated effects of CAR-T is another important cause of cytopenia. This has been added in the revised version.

Discussion

Two points are worth mentioning in the discussion my view. Please add a brief comment:

Time to hematological response after CAR-T is short, which is crucial in patients with AL amyloidosis, especially with advanced cardiac and renal involvement . This has been added to line 353ish as suggested.

Organ dysfunction does not seem to impact the safety profile and outcome (at least in the short-term) of patients with AL amyloidosis treated with CAR-T cell therapy. Thanks for the note. This has been added to lines 366ish

Table 1

Use either “cardiac” or “heart” . This has been corrected as suggested.

Line “organ involvement”, author Landau et al: total number of patients is 7, so information regarding organ involvement is missing in 1 patient. Data in this regard is not fully available on the abstract, however, other is included to denote the lack of data in that 1 case.

Line “Hematologic response”, author “Oliver-Caldes et al.”, delete ORR 100%, as the paper reports data on only one patient . Thanks for the note, this has been corrceted

Line “organ response”, author “Oliver-Caldes et al.”: 55% reduction. Thanks for the note. This has been Fixed in the revised version.

Figure 1

The title of the figure should refer to CAR-T therapy, instead of “BCMA-. Thanks for the note.This has been fixed in the revised version.

Reviewer 2 Report

Comments and Suggestions for Authors

Congratulation on a well written and relevant review.

One comment: in some places the BCMA abbreviation is in reverse order, please go through the manuscript and correct.

Author Response

Reviewer 2

Congratulation on a well written and relevant review.

One comment: in some places the BCMA abbreviation is in reverse order, please go through the manuscript and correct.

 The authors provide a review of the available literature on BCMA-directed therapy in AL amyloidosis patients. The review is overall clear and comprehensive and the conclusions are consistent with the data presented.

Below a few suggestions

Title

I suggest that you replace “BCMA CAR-T” with “BCMA-targeted, or BCMA-directed therapy” since the review does not just describe CAR-T cell therapy. Thanks for the comment.  We have changed the title has been made as suggested

Paragraph 2

Line 69: are associated “with inferior response rates and survival outcomes” The statement here has been attenuated as suggested.

Line 71: “evaluated the addition of daratumumab to the standard of care…” Thanks for the note. We have changed this comment as suggested.

Paragraph 3

Line 94: Add the median prior lines of therapy. Thanks for the note. This has been added in the revised version of the manuscript.

Line 100: I do not fully agree with this statement, as ocular toxicity has been demonstrated to be manageable with schedule modifications with no significant impact on outcomes. Thanks for the comment, we have  changed/added a comment about this as suggested.

Line 113: “developed approved in RRMM” The explanation of RRMM has been provided in the revised version

Line 125: Explicate that the therapeutic target is BCMA. Thanks for the note. This has been corrected

Paragraph 5

Line 144: BMCA BCMA. Thanks for highlighting this typo, this has been corrected in the revised version of the manuscript

Line 146: and

Line 141: “from 3 total case(s)?: do you mean “from 3 case reports”? Thanks for the note. This has been clarified in the revised version.

Line 148: thirteen 13 (use the number, in line with the other data reported). Thanks for the comment, we have modified this to include Thirteen instead.

Line 154: cardiac infiltration involvement. Can you specify how many patients had advanced cardiac involvement? Thanks for the note. Unfortunately, we can’t really do this as some of the reports didn’t explicitly note degree of cardiac involvement.

Line 149: NCX. This has been fixed in the revised version

Line 170: discovered diagnosed on gastric biopsy. Thanks for the note. We have edited this line as suggested.

Line 170: They? Thanks for the note. We have changed to read “the patient”

Line 180: Can you grade CRS, ICANS and infections? Thanks for the note, This has been added in the revised version of the maniuscript

Lines 200-202: Unclear, please rephrase. Thanks for the note, this has been modified in the revised version of the manuscript

Line 201: MRD-negative in the bone marrow 2 . This has been corrected

Reviewer 3 Report

Comments and Suggestions for Authors

Dear Authors,

Please find attached my comments

Author Response

Line 242: Although response rates are quite impressive, the mortality rate is high as well. Is the cause of death reported in the study (infections? organ failure).Thanks for the comment. The study described here showed indeed a higher mortality likely combination of infections and organ involvement. Unfortunately, not much more detail into the deaths is described in the paper.

Paragraph 6

Line 265: grade II-IV grade III-IV CRS? This has been modified in the revised version

Line 296: Immune-mediated effects of CAR-T is another important cause of cytopenia. This has been added in the revised version.

Discussion

Two points are worth mentioning in the discussion my view. Please add a brief comment:

Time to hematological response after CAR-T is short, which is crucial in patients with AL amyloidosis, especially with advanced cardiac and renal involvement . This has been added to line 353ish as suggested.

Organ dysfunction does not seem to impact the safety profile and outcome (at least in the short-term) of patients with AL amyloidosis treated with CAR-T cell therapy. Thanks for the note. This has been added to lines 366ish

Table 1

Use either “cardiac” or “heart” . This has been corrected as suggested.

Line “organ involvement”, author Landau et al: total number of patients is 7, so information regarding organ involvement is missing in 1 patient. Data in this regard is not fully available on the abstract, however, other is included to denote the lack of data in that 1 case.

Line “Hematologic response”, author “Oliver-Caldes et al.”, delete ORR 100%, as the paper reports data on only one patient . Thanks for the note, this has been corrceted

Line “organ response”, author “Oliver-Caldes et al.”: 55% reduction. Thanks for the note. This has been Fixed in the revised version.

Figure 1

The title of the figure should refer to CAR-T therapy, instead of “BCMA-. Thanks for the note.This has been fixed in the revised version.

Round 2

Reviewer 3 Report

Comments and Suggestions for Authors

The authors have adequately addressed most of my comments. I have just a couple of remarks.

Figure 1. Not only the caption, but also the title of the figure should be fixed  (“BCMA CAR-T therapy”, instead of “BCMA-directed therapy”)

Introduction, line 81. The presence of high risk cytogenetic aberrations has probably a limited impact on the treatment choice in patients with relapsed/refractory AL amyloidosis. Main drivers of the choice in this setting are prior regimen(s) and toxicities; type and degree of organ involvement.

Author Response

Response to reviewers:

Thanks for the comments, please see below answers.

Figure 1. Not only the caption, but also the title of the figure should be fixed  (“BCMA CAR-T therapy”, instead of “BCMA-directed therapy”) . This has been modified in the revised version.

Introduction, line 81. The presence of high risk cytogenetic aberrations has probably a limited impact on the treatment choice in patients with relapsed/refractory AL amyloidosis. Main drivers of the choice in this setting are prior regimen(s) and toxicities; type and degree of organ involvement. Thanks for the note, we do try to refer here to the impact of t(11;14) and 1q gains that could represent a more aggressive phenotype and possibility for use of BCL-2 directed treatments at least on the t(11;14) group.